# QuantumIS: A Qualia Consciousness Awareness and Information Theory Quale Approach to Reducing Strategic Decision-Making Entropy

**DOI:** 10.3390/e21020125

**Published:** 2019-01-29

**Authors:** James A. Rodger

**Affiliations:** Eberly College of Business & IT, Indiana University of Pennsylvania, Indiana, PA 15701, USA; jrodger@iup.edu; Tel.: +724-357-5944; Fax: +724-357-4831

**Keywords:** metacognition, learning, competitive advantage, flexibility, expertise, trust, top management support, information system (IS) strategic planning, decision-making, qualia, entropy, quale, consciousness, cognition

## Abstract

This paper investigates the underlying driving force in strategic decision-making. From a conceptual standpoint, few studies empirically studied the decision-maker’s intrinsic state composed of entropy and uncertainty. This study examines a mutual information theory approach integrated into a state of qualia complexity that minimizes exclusion and maximizes the interactions of the information system and its dynamic environment via logical metonymy, illusion, and epigenetics. The article questions whether decision-makers at all levels of the organization are responding from the consciousness of an objective quale from a more subjective qualia awareness in the narrow-sense perspective of individual instances of their conscious experience. To quantify this research question, we explore several hypotheses revolving around strategic information system decisions. In this research, we posit that the eigenvalues of factor analysis along with the reduction in the uncertainty coefficients of the qualia entropy will be balanced by the quale enthalpy of our information theory structural equation model of trust, flexibility, expertise, top management support, and competitive advantage performance. We operationalize the integration of the aforementioned top management support, information systems competencies, and competitive advantage performance concepts into the qualia consciousness awareness and information theory quale framework.

## 1. Introduction

One of the basic principles of quantum mechanics revolves around the energy of the system. In this context, the Hamiltonian is the operator corresponding to the total energy of the system. It is the set of possible outcomes for the total energy of a system and is of fundamental importance in the evolution of quantum theory. In determining the Hamiltonian of a quantum system, eigenvectors are used to determine the different energy eigenstates, and their eigenvalues are the corresponding energy levels used to find the eigenvectors. In this article, we propose that information systems (IS) follow these same principles [1,2,3,4,5]. However, this paper does not assume that IS systems work at equilibrium, nor does it propose that an analogy with the Hamiltonian system theory of entropy production can be drawn. Most organizations’ decision-making takes place in the presence of conflict, paradox, and noise. This study and thesis is a mere hypothesis and we restrict ourselves to the ideal case that an IS system can be considered at equilibrium and that entropy changes can be viewed in this context. We invoke the laws of thermodynamics to propose that organizations and IS respond to similar quantum principles [6,7]). We propose that useful work of the information system can be measured as competitive advantage performance. For example, although one form of energy, such as heat, can be transformed into another form, such as kinetic, potential, chemical, or electrical, the total sum of the energy in any system of materials or bodies remains constant. If some amount of heat (Q) is put into the system, then it must either do work (W′) or increase the total energy of the system. If H stands for the enthalpy of heat content, then ∆H = Q − W′. Therefore, we infer that the total sum of energy for an organization derived from its IS can be neither created nor destroyed but only changes form. For example, the energy derived from IS expertise can be transformed into another form, such as IS infrastructure flexibility, trusting IS–business relationships, IS top management support, or IS competitive advantage performance.

The second law of thermodynamics states that systems tend to reach a state of equilibrium. In other words, any organization, group of individuals, or information system, no matter how highly organized it may be at any given instant, tends toward greater disorder or randomization, called entropy (S). Entropy can be related to the random movements of molecules and can be measured by T∆S, where T is the absolute temperature of the system. When the system is at equilibrium, there is no net reaction and the system has maximum entropy with no capacity to do useful work: Q = T∆S. Work can be done by systems tending toward equilibrium, and a measure of this work is W′ = −∆H + T∆S. Free energy is defined as ∆G = ∆H − T∆S, where ∆G = −W′, such that when the measure of W′ is positive and can do useful work, then the measure of ∆G is negative and vice versa.

A system at equilibrium has three characteristics: There are two opposing tendencies or reactions; both reactions are in full operation at equal speeds, and any change in the conditions, such as concentration, pressure, or temperature, produces a corresponding change in the system. Any change in conditions forces the system to settle down into a new state of equilibrium with new proportions of reactants and products that correspond to these new conditions. In the study of dynamical systems, discrimination of the presence of correlations in time series emerges as one key task. Given a time series, one of the most natural measures of disorder and, thus, the absence of correlation, is Shannon entropy [8], which states that, given a discrete probability distribution, P = {p i:i = 1 to M}. Shannon entropy is defined as follows:*M**S*[*P*] = − ∑ *p i* log (*p i*).*i* = 1(1)

The reduction of uncertainty can be quantified as mutual information (MI). MI can also be described in terms of P2 as MI[P, Q] = H[P] + H[Q] − H[PQ], where Q is another discrete probability distribution [9]. Let X and Y be two random variables; [10] points out that Shannon’s law states that, at the other extreme, if S[P] = log (M), knowledge about the organizational system is minimal, meaning that all states are equally probable (trust or no trust in the IS–business relationship).

Tsuchiya et al. [11] claim that Intrinsic Information Theory provides both a conceptual and a computational structure that is claimed to be identical to experience or qualia and simulated by neural networks [12,13,14,15,16,17,18,19,20]. We postulate that, within a system, consciousness can be measured by the amount of entropy, certainty, and affects that are present. In this manner, we can describe how the components of the system (trust, expertise, flexibility, top management support) causally and irreducibly contribute to the performance and competitive advantage of the whole system. IIT can be used to explain the conscious state of mind of the decision-makers in our study and identifies five essential properties of consciousness [21,22,23]. The first property, intrinsic existence, states that decision-makers experience consciousness only from their intrinsic perspectives [24,25,26,27,28]. Therefore, their consciousness during the completion of a survey instrument exists independently of external observers. The second property is composition, meaning that decision-makers’ consciousness is structured as a moment of experience [29,30]. The third property is information, which can be described as one moment of consciousness that is highly informative [31,32,33,34]. An example would be trust or no trust in IS–business relationships. The fourth property is integration, meaning that it is not possible to experience two independent consciousnesses at the same time [35]. The fifth property is exclusion, meaning that consciousness follows spatial and temporal boundaries [36]. 

For example, if an organization has a trusting IS–business relationship, then it can continue to have trust, or there can be no trust and trust is developed, or there is trust that turns to no trust, or finally, there is no trust that leads to no trust. Uncertainty can be designated as having no trust in the IS–business relationship. Decision-makers are viewing trust, expertise, top management support, and flexibility in the present state. The amount of information lost is known as integrated information. Here, we would look at trust, expertise, top management support, and flexibility in view of performance and competitive advantage. All possible subsets of a system determine the composition and, therefore, we must study the system ceteris paribus, as there are conceivably an infinite number of variables. 

## 2. Theoretical Development of Organizational Competencies

### 2.1. Background

#### 2.2.1. Trust and Infrastructure Competencies

Despite this understanding of the importance of IS competencies in business performance, little research examined how a firm refines IS competencies over time. This article fills this gap in the literature by examining empirically how a firm refines IS competencies such as top management support in the context of competitive advantage performance. Below, we define the concept of IS competencies in this context. Based on this categorization, we define IS competencies as those attributes of IS that cannot be easily imitated by other IS units in different firms [37]. 

#### 2.2.2. IS Infrastructure Flexibility

A flexible IS infrastructure allows for the sharing of data and applications through communication networks. It pertains to arrangements of hardware, software, and networks such that data and applications can be accessed and shared within and between suppliers, customers, and vendors. A flexible IS infrastructure helps in integrating disparate and geographically distributed manufacturing systems and ensures the cost effectiveness of the operation and support of IS applications. Therefore, a flexible infrastructure becomes a critical source of advantage and performance for the firm.

Porto de Albuquerque and Christ [38] studied tensions between business process modeling and IS flexibility, and revealed multiple dimensions using a sociomaterial approach. They investigated the influence of service-oriented architecture (SOA) governance mechanisms on IT flexibility and service reuse. 

#### 2.2.3. Shared IS Business Expertise

A flexible infrastructure with shared IS business expertise is a critical source of advantage and performance for the firm.

Hypothesis 1: IS infrastructure flexibility will have a positive relationship with IS business expertise.

Hypothesis 2: Competitive advantage performance will have a positive relationship with IS expertise.

#### 2.2.4. A Trusting IS–Business Relationship 

Makkonen and Vuori [39] pointed out that a more holistic approach needs to be adopted concerning perceptions of IT and business relationships. In fact, IT has long been acknowledged as an integral part of interorganizational business exchange. Finally, Xu et al. [40] provided guidance to practitioners by enabling them to focus on IS development and training. In this manner, insights were given into how best to prepare IT and customer relationship employees regarding the diverse aspects of trust that are most salient to their business customer needs.

Hypothesis 3: A trusting IS–business relationship will have a positive relationship with competitive advantages.

#### 2.2.5. IS Competitive Advantage Performance

The literature does not provide a consistent picture of payback for IT investment (Lehoux et al., [41]. Their study related chief information officer (CIO) background and attitude toward IT investment to objective measures of a firm’s performance. They found that financial measures tended to be higher when the CIO was from IT rather than general management. However, they also found that an IT manager in a firm with a strategic orientation to IT rather than a utilitarian one was more likely to have better financial performance. Leifer [42] contended that an innovator strategy is associated with a superior firm performance strategy. He believed that, under conditions of environmental dynamism, organizational leaders need to consider the external environments under which their organizations operate and the ability of their IS strategy to impact performance.

#### 2.2.6. IS Top Management Support

Hough and Duffy [43] studied decision support systems at the level of top management. Their study, which found that, in general, respondents were reasonably aware of decision support systems, is useful for addressing what top managers perceive to be their major difficulties in decision-making. In a similar manner, Leisti and Häkkinen [44] posited the importance of introspection on decision-making. Lin et al. [45] indicated that innovative information technologies will create or improve a product and enhance intraorganizational efficiency and effectiveness. They based their research on the framework of upper echelon theory. They investigated the relationships between top managers’ individual differences and IIT championing. Marble [46] reported that the organizational priority given to implementation projects by top management is only indirectly associated with improved user information satisfaction. Only when this priority occurs in the management of continuing development and enhancement does top management support seem to be significant to users. The efficiency and flexibility of the development process was significant in its own right, even without any effects of top management support. They indicated that top management support has implications for IT professionals who acquired different skills and knowledge from their job experiences. 

Hypothesis 4: Competitive advantage performance will have a positive relationship with IS top management support.

Hypothesis 5: IS top management support will have a positive relationship with a trusting IS–business relationship.

Hypothesis 6: IS top management support will have a positive relationship with IS business expertise.

Hypothesis 7: IS top management support will have a positive relationship with IS infrastructure flexibility.

#### 2.2.7. Refining IS Competencies for Business Advantage and Performance

We define the relationship between IS competencies and competitive strategies similarly to Rodger and Bhatt [37]. We base our conceptualization on the literature, cite research, and argue that very little attention is paid to understanding enterprise resource planning (ERP) system usage among adopting firms. We examined the concept of competitive strategy through competitive advantage capability, specifically the impact on ERP system usage. We incorporated user satisfaction to argue that this capability has an indirect effect on user satisfaction, as well as a direct effect on ERP system usage. The integration of the aforementioned top management support, IS competencies, and competitive advantage performance concepts into qualia consciousness awareness (QCA) and information theory quale (ITQ) was captured in the framework seen in Figure 1.

## 3. The Research Model

We posit that, under specific circumstances, constraints, and assumptions, the entropy of the IS can be minimized and the organization performance improved. These specifics are addressed with the needed strongest of supports of increased IS trust, flexibility, business expertise, competitive advantage performance, and top management support. 

We hypothesize that, at the QCA and ITQ levels, flexibility in the IS infrastructure will indirectly impact IS business expertise (see Figure 2) and top management support will be positively related to expertise, trust, and flexibility. In turn, trust will positively affect expertise, which will positively affect competitive advantage performance. We assume that a relationship between top management support and competitive advantage explains how IS competencies can be refined by this relationship. 

## 4. Hypothesis Development

Hu [47] examined both efficiency-centered and novelty-centered business models to investigate how they affect competitive advantage performance. The study looked at how business models affect technological innovation through the mediating role of competitive advantage to help top managers better understand the influence of business models on performance and technological innovation. Boyle’s law states that the volume of a gas at constant temperature is inversely proportional to the pressure to which it is subjected: P1/P2 = V2/V1. In our adaptation, the effects of increasing pressure from top management without concurrent support will lead to a decrease in the volume of the information system and a decrease in the competitive advantage of the organization. This leads to the hypothesis below.

The equilibrium existing in a system may be shifted by increasing the speed of the forward or backward reaction: A + B = C + D. In other words, an increase in the concentration of IS personnel and business partner trust plus top management support leads to increased competitive advantage and improved performance. The equilibrium may also be shifted to the right by the removal of C or D or to the left by the removal of A or B. This leads to the equilibrium constant: [C][D]/[A][B] = K_eq_. If the concentration of A is increased when the system is in equilibrium, then the momentary value of this expression will be smaller and the system must shift to make the expression equal to K_eq_ again. Therefore, we offer the hypothesis below.

Another component of our model states that catalysts cannot shift equilibrium. Therefore, IS expertise is brought alternatively in and out of the overall reaction. Therefore, it follows that any increase in the speed of the forward reaction of IS expertise toward competitive advantage can also increase the speed of the reverse reaction if top management support for expertise is withdrawn. This leads to the hypothesis below.

Increasing the temperature of a reacting system in equilibrium will increase the velocities of the reactions in both directions. When any reaction proceeds in one direction, then heat is either evolved or absorbed. Charles’s law states that volume is directly proportional to absolute temperature: V1/T1 = V2/T2. In a similar manner, we propose that, if top management support increases, then the volume (size) of the IS infrastructure flexibility will proceed toward a greater equilibrium and the organization will evolve toward greater competitive advantage.

## 5. Methodology and Analysis

From the ITQ perspective, data were analyzed using hierarchical regression analysis and structural equation modeling (SEM) to understand the broad objective relationship of top management support with IS competencies and competitive advantage performance. The QCA data were analyzed utilizing both factor analysis eigenvalues and entropy uncertainty coefficients.

### Construct Measurement for Top Management Support, IS Competencies Measures, and Competitive Advantage Performance

We used the following items to measure trusting IS–business relationships: “our business managers and IS personnel share responsibility in setting business strategy”; “our business managers and IS personnel jointly set business strategy”; “our business managers and IS personnel appreciate each other’s contributions in setting business strategy”; “our business managers and IS personnel periodically consult each other in setting business objectives”; and “our business managers and IS personnel trust each other in setting business objectives”.

Finally, we used the following items to measure competitive advantage: “employees are important to competitive advantage performance”; “external reports are important to competitive advantage performance”; “competitors are important to competitive advantage performance”; and “top managers seek outside information to competitive advantage performance.”

## 6. Data Analysis and Results

### 6.1. Structural Equation Model for ITQ

Given the multiple relationships in our model, we used LISREL for our quale data analysis. One of its primary strengths is its simultaneous estimation of the measurement and structural models. We used SEM to test our hypotheses (see Figure 2).

Figure 3 shows that all five correlations were significant. Because this correlation between flexibility, expertise, trust, top management support, and competitive advantage performance was previously hypothesized, we can establish that the hypotheses are accepted and that the latent variables flexibility, expertise, trust, and top management support are correlated with competitive advantage performance. Overall, the model demonstrated adequate fit to the data: χ^2^ = 237, degrees of freedom (df) = 94, comparative fit index = 0.933, goodness-of-fit index = 0.960, normed fit index = 0.895, root-mean-square error of approximation = 0.0671, *p* ≤ 0.00452. Our results supported all seven hypotheses: Hypothesis 1, IS infrastructure flexibility will have a positive relationship with IS business expertise (B = 0.67); Hypothesis 2, competitive advantage performance will have a positive relationship with IS expertise (B = 0.58); Hypothesis 3, IS business expertise will have a positive relationship with a trusting IS–business relationship (B = 0.29); Hypothesis 4, competitive advantage performance will have a positive relationship with IS top management support (B = 0.77); Hypothesis 5, IS top management support will have a positive relationship with a trusting IS–business relationship (B = 0.34); Hypothesis 6, IS top management support will have a positive relationship with IS business expertise (B = 0.63); and Hypothesis 7, IS top management support will have a positive relationship with IS infrastructure flexibility (B = 0.49).

### 6.2. Reliability and Regression Analysis

Table 1 shows an *R*^2^ of 0.415; this means that nearly 42% of the variance in the dependent variable (competitive advantage performance) was explained by the independent variable flexibility of the IS infrastructure, top management support, IS business expertise, and trust. Table 2 demonstrates that the relationship between these variables was significant at *p* ≤ 0.000. Table 3 gives the individual contributions. For example, Flexibility 1 and 3 had a significance of *p* ≤ 0.01. Expertise 2 and 3 had significances of *p* ≤ 0.001 and *p* ≤ 0.002, respectively. Trust 3 had *p* ≤ 0.001, and top management support 1, 2, and 3 all had *p* ≤ 0.000.

LISREL was used for our quale data analysis. We checked the measures by assessing their reliability and unidimensionality. We iteratively revised the measurement model by dropping, one at a time, the items that shared a high degree of residual variance with other items. Cronbach’s alpha for all 17 items was 0.855, as illustrated in Table 4. Table 5 provides item-level correlations. The correlations between all variables were good, and these relationships were strengthened by the SEM results in Figure 2. Table 6 gives means and standard deviations for construct items. The results of an exploratory factor analysis for all items also supported the unidimensionality of the constructs, as shown in Table 7. Table 7 shows the results of principal component analysis with varimax rotation. The results of the exploratory factor analysis for all items supported the unidimensionality of the constructs.

### 6.3. Descriptive Statistics, Discriminant Validity, and Convergent Validity

Table 8 shows the overall descriptive statistics. Means ranged from 4.47 for the composite competitive advantage to 5.16 for trust. Standard deviations ranged from 0.522 for the competitive advantage composite to 0.769 for composite support. Convergent validity is supported by the results in Table 9. The standardized factor loadings of measurement items on their respective factors were all significant. Composite reliability ranged from 0.420 to 0.788. Table 10 demonstrates that the test of between-subject effects of the overall composite model was significant at *p* = 0.000. The interactive effects of expertise, infrastructure, and trust contributed significantly to competitive advantage, and the least-squares regression was weighted by top management support. To assess discriminant validity, we set some regression weights to 1 and did not estimate them to determine whether their correlations were significantly different from unity. When competitive advantage increased by 1, expertise 1 increased by 1 as well. The same was true for competitive advantage on top management support 1. The probability of obtaining a critical ratio as large as the absolute value was less than 0.0001 for most of the regression weights. The regression weight for the latent variable competitive advantage in the prediction of expertise 2 (0.522 ***) was significantly different from 0 at the 0.000 level. The same was true for the regression weight for competitive advantage on top management support 2 (0.661 ***). The standard error shows how accurately the values of the free parameters were estimated. The standard error was small, 0.08 (competitive advantage); therefore, we believe that the parameters were estimated correctly. For each free parameter, the parameter estimate divided by its standard error produces a *t*-value. If a *t*-value is between –1.96 and 1.96, it is not significantly different from zero; thus, fitting it to zero will not make the fit of the model significantly worse. For example, the relationship between competitive advantage and expertise had an estimate of 0.58 ** and a standard error of 0.08.

### 6.4. QCA Qualia Uncertainty Coefficient Measurements of Entropy

Table 11 looks at the IS flexibility variable. Lambda measures the percentage of error reduction when the independent variable is used to predict the dependent variable. Calculation based on any desired outcome contributing to lambda assumes that lambda ranges from 0 to 1. As usual, we want significance <0.050 for lambda to be statistically significant. Because competitive advantage performance was dependent, flexibility was a significant predictor. Flexibility contributed 42.1% ± 3.89 (Standard Error) of the variability of competitive advantage performance. In addition, competitive advantage performance was a good predictor of flexibility (significance = 0.000). The symmetric value was significant, and its value was between the other two lambda values. Goodman and Kruskal’s (1954) tau is similar to lambda but is based on predictions in the same proportion as the marginal totals (individual row or column subtotals). No symmetric value is given because it is only directional and it predicts that both variables can be significant. The uncertainty (entropy) coefficient is a measure of association that indicates the proportional reduction in error when values of one variable are used to predict values of the other variable; both symmetric and directional versions are calculated. The proportional reduction in error was 55.2%. In other words, having a flexible information system reduced the entropy or probability that we would make a prediction error on the competitive advantage performance dependent variable. Alternatively, having access to the flexibility variable improved our probability of predicting the correct competitive advantage performance level by 55.2%. Because the uncertainty coefficient uses the entire distribution of data to draw its conclusions, it follows that it is a good measure for reducing IS flexibility entropy.

Table 12 looks at the IS personnel expertise variable. Because competitive advantage performance was dependent, expertise was a significant (significance = 0.000) predictor. Expertise contributed 37.4% ± 4.00 (SE) of the variability of competitive advantage performance. In addition, competitive advantage performance was a good predictor of expertise (significance = 0.000). The symmetric value was significant, and its value was between the other two lambda values. The proportional reduction in error was 56.7%. In other words, having information system expertise reduced the entropy or probability that we would make a prediction error on the competitive advantage performance dependent variable. Alternatively, having access to the expertise variable improved our probability of predicting the correct competitive advantage performance level by 56.7%. Because the uncertainty coefficient uses the entire distribution of data to draw its conclusions, it follows that it is a good measure for reducing personnel expertise entropy.

Table 13 looks at the IS–business relationship trust variable. Because competitive advantage performance was dependent, trust was a significant (significance = 0.000) predictor. Trust contributed 23.6% ± 3.20 (SE) of the variability of competitive advantage performance. In addition, competitive advantage performance was a good predictor of trust (significance = 0.000). The symmetric value was significant, and its value was between the other two lambda values. The proportional reduction in error was 38.3%. In other words, having personnel trust reduced the entropy or probability that we would make a prediction error on the competitive advantage performance dependent variable. Alternatively, having access to the trust variable improved our probability of predicting the correct competitive advantage performance level by 38.3%. Because the uncertainty coefficient uses the entire distribution of data to draw its conclusions, it follows that it is a good measure for reducing personnel trust entropy.

Table 14 looks at the top management support variable. Because competitive advantage performance was dependent, trust was a significant (significance = 0.000) predictor. Support contributed 54.3% ± 3.30 (SE) of the variability of competitive advantage performance. In addition, competitive advantage performance was a good predictor of support (significance = 0.000). The symmetric value was significant, and its value was between the other two lambda values. The proportional reduction in error was 63.5%. In other words, top management support reduced the entropy or probability that we would make a prediction error on the competitive advantage performance dependent variable. Alternatively, having access to the support variable improved our probability of predicting the correct competitive advantage performance level by 63.5%. Because the uncertainty coefficient uses the entire distribution of data to draw its conclusions, it follows that it is a good measure for reducing top management support entropy.

The initial factor analysis report in Table 15 shows eigenvalues obtained from a principal component analysis. The scree plot graphs these eigenvalues. The number of factors suggested is equal to the number of eigenvalues that exceed 1.0. Alternatively, the scree plot can be used to guide the initial choice of the number of factors. The number of eigenvalues that appear before the scree plot levels out can provide an upper bound on the number of factors. The scree plot shown in Figure 4 begins to level out after the fourth eigenvalue. Table 15 indicates that the first eigenvalue accounted for 75.98% of the variance and the second eigenvalue accounted for 18.69%, for a total of 94.67% of the total variance. The third eigenvalue explained only 4.02% of the variance, and the contributions from the remaining eigenvalues were 1.19% and 0.121%. Although the number of factors was initially set to 1, this analysis suggested that extracting five factors was appropriate, as shown in Table 16.

Table 17 summarizes the link between our proposed ITQ MI and the statistical tools used in the manuscript: correlations, Kendall coefficients, ANOVA, and factor analysis results. A clear and concise section explaining the statistical methodologies used, and the relationships between them (step by step) is provided below. ITQ is composed of five components. These are the state, uncertainty, mutual information integration, and complexity of the collective consciousness. This decision-making is contrasted with the individual consciousness or qualia components which include intrinsic, composition, information, integration, and exclusion elements. We posit that, under specific circumstances, constraints, and assumptions, the entropy of the IS can be minimized and the organization performance of both individual and collective decision-making improved. These specifics are addressed with the needed strongest of supports of increased IS trust, flexibility, business expertise, competitive advantage performance, and top management. Therefore, a regression analysis was run in order to demonstrate this potential relationship, along with the individual contributions of these decision-making supporters and their reliability for internal consistency purposes. The descriptive statistics and correlations supported these findings. Factor analysis was run in order to develop the supporters and demonstrate their interconstruct correlations. In order to compare and contrast individual and collective entropy, directional measures such as Kendall coefficients and structured equation modeling were respectively employed to validate our model.

## 7. Discussion

Our ITQ quale results generally supported the validity of our decision-makers’ broad objective contribution to the validity of our research model. We showed, from a quale perspective, that top management support enhanced IT infrastructure flexibility, a trusting IS–business relationship, IS business expertise, and competitive advantage performance. In turn, IS business expertise increased competitive advantage performance. Top management support generated IS competencies, and this directly impacted the firm’s ability to improve competitive advantage performance. We showed that the flexibility of the IT infrastructure is a critical organizational IT capability for improving competitive advantage performance. In other words, IT infrastructure flexibility is a capability that has strategic ramifications and provides top management with the ability to respond rapidly to critical requirements for firm survival and success. In a similar manner, trust between IS personnel and their business partners, as well as IS expertise to run flexible IS, all benefit competitive advantage performance through the support of top management. We showed that top management support coupled with IT infrastructure flexibility, a trusting IS–business relationship, and IS expertise enabled organizations to perform at a higher level through the competitive advantage performance that they generated.

Through the QCA qualia results, we were able to show that decision-makers conveyed, through their conscious awareness, that they were able to lower the entropy of the organization when information needs changed, and that they were able to reconfigure and adapt their IT infrastructure according to their information-processing needs. Decision-makers’ individual subjective narrow awareness in their consciousness is instrumental in generating relevant and timely information in the ITQ sense so that firms are more likely to develop competitive advantages to improve their performance. By following this thought process, top management decision-makers are better equipped to exploit existing competencies of trust, expertise, and flexibility to generate long-term performance opportunities for competitive advantage.

## 8. Limitations

Although our study used the same respondents to provide data on both our independent and dependent variables, we attempted to overcome this common method bias by both measuring a broad objective ITQ quale approach and balancing it with the narrower subjective QCA qualia consciousness of the decision-makers. The combined results indicate that this bias did not seem to be a major problem. However, we recognize that it still may have biased our results. Another limitation that we tried to overcome was the static cross-sectional snapshot of top management support, capabilities, and competitive advantage. Although this approach makes it difficult to address the interaction of these issues, we built on the data and findings of a previous similar study (Bhatt et al., 2010) to improve the longitudinal perspective of how top management support, IT capabilities, and trust lead to competitive advantage performance that is created over time.

## 9. Conclusions

Our ITQ SEM quale approach showed that IT infrastructure flexibility, top management support, trust, and expertise can generate competitive advantage and, thus, increase a firm’s ability to leverage performance enhancement. Top management is able to funnel its subjective QCA consciousness into more broad-based objective ITQ information to respond to market opportunities, thereby creating improvements in competitive advantage and firm performance. For example, our study showed that it is important that decision-makers overcome their subjective narrow QCA qualia to lower the entropy of the organization to develop IT infrastructures that can facilitate the leveraging of competitive advantage performance by the organization. Our study showed the importance of the intersection of QCA qualia consciousness with ITQ and the value of decreasing entropy within the organization as demonstrated in Figure 1. For example, ITQ was used in the context of reducing expertise uncertainty in the intrinsic QCA of the decision-maker. In a similar manner, trust ensues if ITQ mutual information is paired with the QCA information of the individual decision-maker. QCA composition of IS flexibility leads to ITQ integration in the mind of the top managers and their actions to support the ITQ of mutual information for performance [35]. We also observed that excluding complexity is an important factor in evaluating the entropy within an organization. Future studies may wish to concentrate on the blank areas in Figure 1 to shed more light on these interactions between QCA and ITQ.

Although entropy can describe globally the level of disorder of a process, there is also the probability that an analysis of time series using solely Shannon entropy could be incomplete. The reason for this is that an entropy measure does not quantify the degree of structure or patterns present in a process. Consequently, we used our framework to propose that ITQ statistical complexity is necessary to characterize a system and explore the QCA mindset of decision-makers. To observe more clearly the temporal changes in ITQ informational efficiency, we computed the uncertainty coefficient and used eigenvalues from factor analysis to measure entropy in all of the QCA decision variables studied. We hypothesized that if the value of the Shannon S interpretation is 2/3lnk < S ≤ lnk, then the organizational system has a high level of disorder, entropy, and uncertainty. However, if 1/3lnk < S ≤ 2/3lnk, then the organizational system is somewhat ordered and we should determine whether any of the factors should be strengthened. Finally, if 0 < S ≤ 1/3lnk, then the organizational system is highly ordered, with certainty and enthalpy, and the variables used to measure organizational sustainability are grouped around a common low uncertainty, high efficiency, and low entropy point.

In other words, we adopted a variation in the information theory approach that involves the main concepts of entropy and uncertainty. These properties permit a consideration of entropy as a measure of the uncertainty of random QCA variables and provides the theoretical basis for measuring our model ITQ information. As our ITQ information increases, our knowledge of our organizational model decreases the degree of QCA uncertainty. By utilizing the concept of entropy, we can precisely measure the degree of uncertainty of the QCA random variables describing the ITQ information relationship between trust, expertise, flexibility, and competitive advantage in our organizational system. For example, trust and infrastructure competencies (IS competencies) were an important area of inquiry for both top managers and academicians. It is now widely believed that, to achieve a sustainable competitive advantage, a firm must be able to renew IS personnel, IS business trust, and infrastructure flexibility. Although the literature discussed the importance of competitive advantage through IS, little is known about how top management support renews IS competencies and helps in achieving competitive advantage. Even less is known about trust and the organization’s ability to form a relationship between IS groups and business groups so that these groups can work together to solve organization-wide problems. This is considered a critical attribute of IS competencies.

In this research, we took a step in this direction by analyzing the relationship between IS competencies for business advantages and investigating IS infrastructure, expertise, trust, and top management support. We collected data through a survey of 339 IS and business managers from service and manufacturing participants. We found that top management support was positively related to IT infrastructure flexibility and trust from both an ITQ quale and a QCA qualia perspective. Moreover, IS infrastructure flexibility and trust were both positively related to competitive advantage performance. Finally, IS business expertise was positively related to competitive advantage performance. These results show the importance of top management support for refining IS trust and infrastructure competencies. Our results generally support the validity of the proposed research model. Top management support enhances IS flexibility infrastructure, IS business expertise, and IS trust relationships, as well as impacts competitive advantage. Furthermore, IS business expertise has a positive relationship with competitive advantage. We conclude that top management support has an impact on IS capabilities and competitive advantage performance. We showed that it is important that top managers develop scalable, modular, and compatible IT infrastructures that can lead to competitive advantage through trust and expertise. Most important, we validated many of the findings of the Bhatt et al.’s (2010) study, leading to a longitudinal inroad into understanding these constructs through the adoption of our QCA ITQ framework. Survey items can be found in Appendix A and demographics in Appendix B. 

## Figures and Tables

**Figure 1 entropy-21-00125-f001:**
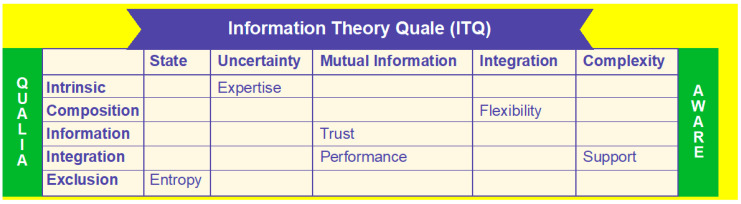
Qualia Consciousness and information theory quale integration with organizational competences.

**Figure 2 entropy-21-00125-f002:**
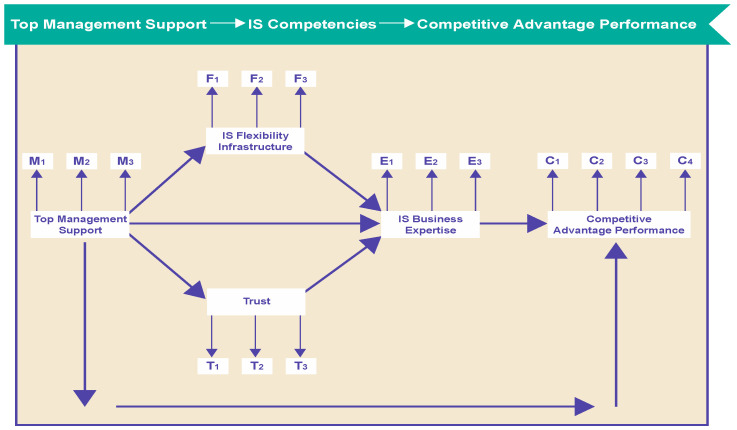
Research model.

**Figure 3 entropy-21-00125-f003:**
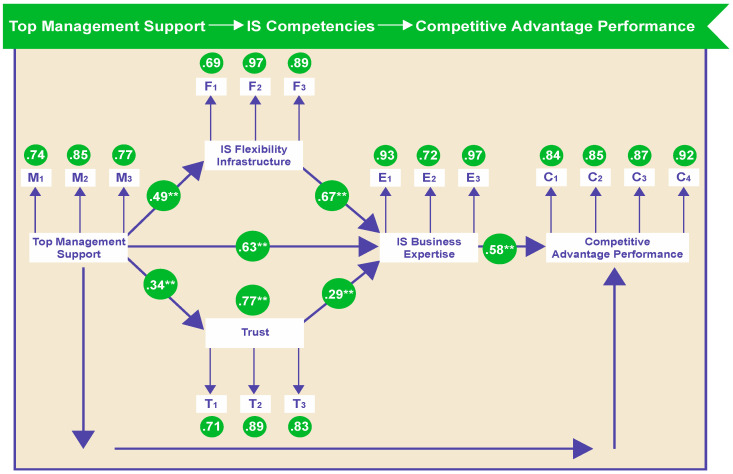
Nomological network of relationships between top management support, IS competencies and competitive advantate structural model results.

**Figure 4 entropy-21-00125-f004:**
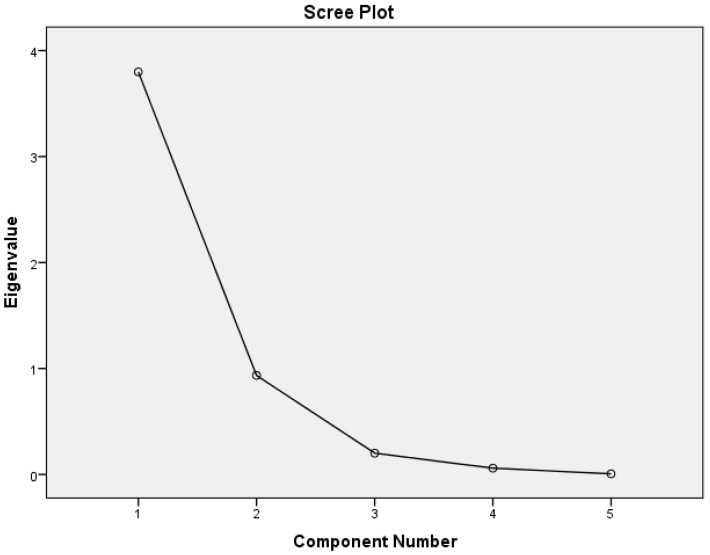
Scree Plot.

**Table 1 entropy-21-00125-t001:** Model summary.

Model	*R*	*R* ^2^	Adjusted *R*^2^	Standard Error of the Estimate
1	0.645 ^a^	0.415	0.394	0.40744

^a^ Predictors: (Constant), support 3, Flex3, Trust3, support2, Flex1, Trust1, support1, Exp2, Trust2, Exp1, Exp3, Flex2.

**Table 2 entropy-21-00125-t002:** ANOVA ^a^; df—degrees of dreedom.

Model		Sum of Squares	df	Mean Square	F	Significance
**1**	Regression	38.216	12	3.185	19.184	0.000 ^b^
	Residual	53.786	324	0.166		
	Total	92.001	336			

^a^ Dependent variable: total CA. ^b^ Predictors: (Constant), support3, Flex3, Trust3, support2, Flex1, Trust1, support1, Exp2, Trust2, Exp1, Exp3, Flex2.

**Table 3 entropy-21-00125-t003:** Coefficients ^a^.

Model		Unstandardized Coefficients		Standardized Coefficients	t	Significance
		B	Standard Error	Beta		
**1**	(Constant)	3.763	0.264		14.274	0.000
	Flex1	−0.094	0.039	−0.140	−2.388	0.018
	Flex2	−0.015	0.064	−0.022	−0.230	0.818
	Flex3	−0.190	0.039	−0.278	−4.863	0.000
	Exp1	0.043	0.069	0.059	0.624	0.533
	Exp2	0.164	0.050	0.276	3.254	0.001
	Exp3	−0.157	0.049	−0.271	−3.194	0.002
	Trust1	0.005	0.044	0.009	0.122	0.903
	Trust2	−0.079	0.056	−0.110	−1.412	0.159
	Trust3	0.163	0.048	0.242	3.362	0.001
	support1	0.370	0.057	0.563	6.501	0.000
	support2	−0.164	0.043	−0.231	−3.794	0.000
	support3	0.093	0.024	0.218	3.842	0.000

^a^ Dependent variable: total CA.

**Table 4 entropy-21-00125-t004:** Reliability statistics.

Cronbach’s Alpha	Number of Items
0.855	17

**Table 5 entropy-21-00125-t005:** Correlations.

	F1	F2	F3	E1	E2	E3	T1	T2	T3	M1	M2	M3	CA1	CA2	CA3	CA4
F1	1	0.629 **	0.378 **	0.501 **	0.519 **	0.565 **	0.457 **	0.431 **	0.422 **	0.359 **	0.116 *	0.330 **	−0.400 **	0.320 **	0.321 **	0.386 **
F2		1	0.511 **	0.758 **	0.661 **	0.656 **	0.726 **	0.501 **	0.541 **	0.331 **	0.199 **	0.330 **	−0.400 **	0.320 **	0.321 **	0.386 **
F3			1	0.566 **	0.500 **	0.529 **	0.433 **	0.171 **	0.043	0.172 **	0.123 *	0.192 **	−0.440 **	0.051	0.015	0.246 **
E1				1	0.717 **	0.655 **	0.667 **	0.396 **	0.264 **	0.432 **	0.294 **	0.273 **	−0.369 **	0.322 **	0.314 **	0.396 **
E2					1	0.782 **	0.515 **	0.536 **	0.393 **	0.469 **	0.443 **	0.310 **	−0.428 **	0.468 **	0.334 **	0.506 **
E3						1	0.545 **	0.602 **	0.404 **	0.579 **	0.394 **	0.430 **	−0.520 **	0.494 **	0.371 **	0.518 **
T1							1	0.333 **	0.462 **	0.189 **	0.232 **	0.251 **	−0.310 **	0.250 **	0.257 **	0.212 **
T2								1	0.654 **	0.739 **	0.430 **	0.517 **	−0.383 **	0.556 **	0.586 **	0.702 **
T3									1	0.490 **	0.338 **	0.434 **	−0.266 **	0.537 **	0.629 **	0.535 **
M1										1	0.589 **	0.555 **	−0.436 **	0.793 **	0.652 **	0.815 **
M2											1	0.482 **	−0.397 **	0.585 **	0.362 **	0.492 **
M3												1	−0.321 **	0.531 **	0.571 **	0.565 **
CA1													1	−0.360 **	−0.148 **	−0.336 **
CA2														1	0.707 **	0.732 **
CA3															1	0.682 **
CA4																1

** significant at 0.01; * Significant at 0.05.

**Table 6 entropy-21-00125-t006:** Descriptive statistics.

	*N*	Mean	Standard Deviation
Flex1	339	4.9971	0.77878
Flex2	339	5.1445	0.78053
Flex3	339	4.9410	0.76309
Exp1	339	5.1062	0.71372
Exp2	339	5.0590	0.87844
Exp3	337	4.9436	0.90292
Trust1	339	5.1858	0.86567
Trust2	339	5.1681	0.73262
Trust3	339	5.1268	0.77602
support1	339	5.1121	0.79522
support2	339	5.4159	0.73451
support3	339	4.8378	10.22846
CA1	339	2.4277	10.55470
CA2	339	5.2419	0.75008
CA3	339	5.1239	0.64081
CA4	339	5.0501	0.81797
Total CA	339	4.4609	0.52220
Valid *N* (listwise)	337		

**Table 7 entropy-21-00125-t007:** Factor analysis.

	Component				
	1	2	3	4	5
F1	0.851	0.081	0.269	0.096	−0.023
F2	0.848	0.265	0.535	0.197	0.006
F3	0.921	0.249	−0.108	0.111	−0.052
E1	0.2180.	0.917	0.165	0.363	−0.123
E2	−0.046	0.858	0.103	−0.020	−0.067
E3	0.211	0.838	0.122	0.111	0.072
T1	0.365	0.069	0.850	0.288	0.135
T2	0.421	0.638	0.906	0.093	0.031
T3	0.555	−0.296	0.875	0.037	−0.029
S1	0.000	0.079	−0.078	0.877	0.150
S2	−0.200	0.474	0.271	0.833	−0.127
S3	0.601	0.157	0.327	0.914	0.160
C1	0.300	−0.043	0.078	0.352	0.856
C2	0.480	0.111	0.192	0.248	0.881
C3	0.232	0.325	0.076	0.138	0.878
C4	0.142	−0.003	−0.142	0.239	0.913

**Table 8 entropy-21-00125-t008:** Descriptive statistics.

	Mean	Standard Deviation	*N*
Flexibility	5.0275	0.63424	339
Expertise	5.0386	0.75165	337
Trust	5.1603	0.63913	339
Support	5.1219	0.76894	339
Competitive	4.4609	0.52220	339

**Table 9 entropy-21-00125-t009:** Interconstruct correlations.

		Flexibility	Expertise	Trust	Support	Competitive
Flexibility	Pearson’s correlation	1	0.788 **	0.636 **	0.373 **	−0.015
	Significance (2-tailed)		0.000	0.000	0.000	0.789
	*N*	339	337	339	339	339
Expertise	Pearson’s correlation	0.788 **	1	0.665 **	0.528 **	0.133 *
	Significance (2-tailed)	0.000		0.000	0.000	0.014
	*N*	337	337	337	337	337
Trust	Pearson’s correlation	0.636 **	0.665 **	1	0.584 **	0.311 **
	Significance (2-tailed)	0.000	0.000		0.000	0.000
	*N*	339	337	339	339	339
Support	Pearson’s correlation	0.373 **	0.528 **	0.584 **	1	0.420 **
	Significance (2-tailed)	0.000	0.000	0.000		0.000
	*N*	339	337	339	339	339
Competitive	Pearson’s correlation	−0.015	0.133 *	0.311 **	0.420 **	1
	Significance (2-tailed)	0.789	0.014	0.000	0.000	
	*N*	339	337	339	339	339

** Correlation is significant at the 0.01 level (2-tailed). * Correlation is significant at the 0.05 level (2-tailed). Dependent variable: competitive.

**Table 10 entropy-21-00125-t010:** Tests of between-subject effects ^a^.

Dependent Variable: Competitive						
Source		Type III Sum of Squares	df	Mean Square	F	Significance
Intercept	Hypothesis	2478.473	1	2478.473	2166.344	0.000
	Error	72.789	63.622	1.144 ^b^		
Expertise	Hypothesis	20.003	6	3.334	4.154	0.000
	Error	251.220	313	0.803 ^c^		
Trust	Hypothesis	19.877	8	2.485	3.096	0.002
	Error	251.220	313	0.803 ^c^		
Flexibility	Hypothesis	14.446	8	1.806	2.250	0.024
	Error	251.220	313	0.803 ^c^		

^a^ Weighted least-squares regression—weighted by support. ^b^ 136 MS(Trust) + 0.113 MS(Flexibility) + 0.752 MS. ^c^ MS.

**Table 11 entropy-21-00125-t011:** Directional measures.

		Value	Asymptotic Standardized Error ^a^	Approximate T ^b^	Approximate Significance
**Lambda**	Symmetric	0.479	0.032	12.383	0.000
	Total flex Dependent	0.553	0.040	11.004	0.000
	Total performance compadv Dependent	0.421	0.038	9.644	0.000
Goodman and Kruskal tau	total flex Dependent	0.551	0.021		0.000^c^
	total performance compadv Dependent	0.325	0.019		0.000 ^c^
Uncertainty Coefficient	Symmetric	0.621	0.017	23.223	0.000 ^d^
	total flex Dependent	0.708	0.018	23.223	0.000 ^d^
	total performance compadv Dependent	0.552	0.018	23.223	0.000 ^d^

^a^ Not assuming the null hypothesis. ^b^ Using the asymptotic standard error assuming the null hypothesis. ^c^ Based on chi-square approximation. ^d^ Likelihood ratio chi-square probability.

**Table 12 entropy-21-00125-t012:** Directional measures.

		Value	Asymptotic Standardized Error ^a^	Approximate T ^b^	Approximate Significance
Lambda	Symmetric	0.457	0.029	12.710	0.000
	Total expertise Dependent	0.558	0.037	12.405	0.000
	Total performance compadv Dependent	0.374	0.040	8.113	0.000
Goodman and Kruskal tau	Total expertise Dependent	0.543	0.022		0.000 ^c^
	Total performance compadv Dependent	0.308	0.017		0.000 ^c^
Uncertainty Coefficient	Symmetric	0.623	0.016	25.576	0.000 ^d^
	Total expertise Dependent	0.691	0.019	25.576	0.000 ^d^
	Total performance compadv Dependent	0.567	0.015	25.576	0.000 ^d^

^a^ Not assuming the null hypothesis. ^b^ Using the asymptotic standard error assuming the null hypothesis. ^c^ Based on chi-square approximation. ^d^ Likelihood ratio chi-square probability.

**Table 13 entropy-21-00125-t013:** Directional measures.

		Value	Asymptotic Standardized Error ^a^	Approximate T ^b^	Approximat Significance
Lambda	Symmetric	0.340	0.032	9.468	0.000
	Total trust Dependent	0.459	0.043	8.885	0.000
	Total performance compadv Dependent	0.236	0.032	6.971	0.000
Goodman and Kruskal tau	Total trust Dependent	0.382	0.022		0.000 ^c^
	Total performance compadv Dependent	0.171	0.012		0.000 ^c^
Uncertainty Coefficient	Symmetric	0.440	0.019	17.632	0.000 ^d^
	Total trust Dependent	0.517	0.023	17.632	0.000 ^d^
	Total performance compadv Dependent	0.383	0.018	17.632	0.000 ^d^

^a^ Not assuming the null hypothesis. ^b^ Using the asymptotic standard error assuming the null hypothesis. ^c^ Based on chi-square approximation. ^d^ Likelihood ratio chi-square probability.

**Table 14 entropy-21-00125-t014:** Directional measures.

		Value	Asymptotic Standardized Error ^a^	Approximate T ^b^	Approximate Significance
**Lambda**	Symmetric	0.562	0.033	14.282	0.000
	Top manage Dependent	0.585	0.040	11.114	0.000
	Total performance compadv Dependent	0.543	0.033	14.315	0.000
Goodman and Kruskal tau	Top manage Dependent	0.576	0.021		0.000 ^c^
	Total performance compadv Dependent	0.439	0.022		0.000^c^
Uncertainty Coefficient	Symmetric	0.680	0.017	25.489	0.000 ^d^
	Top manage Dependent	0.732	0.018	25.489	0.000 ^d^
	Total performance compadv Dependent	0.635	0.018	25.489	0.000 ^d^

^a^ Not assuming the null hypothesis. ^b^ Using the asymptotic standard error assuming the null hypothesis. ^c^ Based on chi-square approximation. ^d^ Likelihood ratio chi-square probability.

**Table 15 entropy-21-00125-t015:** Total variance explained.

Component	Initial Eigenvalues			Extraction Sums of Squared Loadings			Rotation Sums of Squared Loadings		
	Total	% of Variance	Cumulative %	Total	% of Variance	Cumulative %	Total	% of Variance	Cumulative %
1	3.799	75.978	75.978	3.799	75.978	75.978	2.829	56.581	56.581
2	0.935	18.691	94.669	0.935	18.691	94.669	1.132	22.630	79.211
3	0.201	4.022	98.691	0.201	4.022	98.691	0.967	19.338	98.549
4	0.059	1.188	99.879	0.059	1.188	99.879	0.065	1.301	99.850
5	0.006	0.121	100.000	0.006	0.121	100.000	0.007	0.150	100.000

Extraction method: principal component analysis.

**Table 16 entropy-21-00125-t016:** Component transformation matrix.

Component	1	2	3	4	5
1	0.839	0.289	0.460	0.038	0.011
2	−0.365	0.927	0.079	0.035	0.007
3	0.404	0.234	−0.884	0.017	−0.025
4	0.025	0.048	0.009	−0.991	−0.122
5	−0.006	−0.001	0.027	0.122	−0.992

Extraction method: principal component analysis. Rotation method: varimax with Kaiser normalization.

**Table 17 entropy-21-00125-t017:** Qualia consciousness and information theory quale integration with organizational competence.

		State	Uncertainty	Mutual	Integration	Complexity	
Q			Expertise	Information			A
U	Intrinsic				Flexibility		W
A	Composition						A
L	Information			Trust		Support	R
I	Integration			Performance			E
A	Exclusion	Entropy

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
