# Peer review of "QuantumIS: A Qualia Consciousness Awareness and Information Theory Quale Approach to Reducing Strategic Decision-Making Entropy"

_entropy, 2019, doi:10.3390/e21020125_

Round 1
Reviewer 1 Report
In addition to minor typos like the Shannon formula in line 67-69
(the M is displaced), the paper starts by assuming that IS systems work
at equilibrium and that an analogy with Hamiltonian system theory of
entropy production can be drawn. This is a completely wrong claim. Most
organizations' decision making is taking place in the presence of
conflict, paradox and noise. The authors should rephrase their studies
and thesis as a mere hypothesis i.e. restrict themselves in the ideal
case that an IS system can be considered at equilibrium. Under what
specific circumstances? Under which specific constraints? Under which
assumptions? All these questions should be addressed with the needed
strongest of supports. The paper is confusing it needs serious and major
revision taking into account the latest of system theory (see for
example https://en.wikipedia.org/wiki/ Open_system_(systems_theory) )
and consciousness studies (e.g. the recent "The Blackwell Companion to
Consciousness", by Susan Schneider and Max Velmans, for the issue of qualia).
They should also try to limit the number of pages of the main article
by moving all details of descriptive statistics to an Appendix. As is it
adds to the overall confusion in style and in context.
Author Response
Reviewer One Comments and Suggestions for Authors
In addition to minor typos like the Shannon formula in line 67-69 (the M is displaced),
M
S[ P ] = − ∑ p i log (p i )
i =1
The displaced M has been fixed and minor typos addressed.
the paper starts by assuming that IS systems work at equilibrium and that an analogy with Hamiltonian system theory of entropy production can be drawn. This is a completely wrong claim. Most organizations' decision making is taking place in the presence of conflict, paradox and noise. The authors should rephrase their studies and thesis as a mere hypothesis i.e. restrict themselves in the ideal case that an IS system can be considered at equilibrium.
However, this paper does NOT assume that IS systems work at equilibrium nor does it propose that an analogy with Hamiltonian system theory of entropy production can be drawn. Most organizations' decision making takes place in the presence of conflict, paradox and noise. This study and thesis are a mere hypothesis and we restrict ourselves to the ideal case that an IS system can be considered at equilibrium and that Entropy changes can be viewed in this context.
Under what specific circumstances? Under which specific constraints? Under which assumptions? All these questions should be addressed with the needed strongest of supports.
We posit that under specific circumstances, constraints and assumptions the Entropy of the IS can be minimized and the organization performance improved. These specifics are addressed with the needed strongest of supports of increased IS trust, flexibility, business expertise, competitive advantage performance, and top management support.
The paper is confusing it needs serious and major revision taking into account the latest of system theory (see for example https://en.wikipedia.org/wiki/ Open_system_(systems_theory) ) and consciousness studies (e.g. the recent "The Blackwell Companion to Consciousness", by Susan Schneider and Max Velmans, for the issue of qualia).
Thank you for these valuable suggestions. The following references to consciousness, qualia and system theory have been added to the manuscript:
Velmans, M. and Susan Schneider Editor(s): "The Blackwell Companion to Consciousness" 13 December 2007, Blackwell Publishing Ltd.
Maria Charina, Costanza Conti, Mariantonia Cotronei, Mihai Putinar. System theory and orthogonal multi-wavelets. Journal of Approximation Theory, Volume 238, February 2019, Pages 85-102
They should also try to limit the number of pages of the main article by moving all details of descriptive statistics to an Appendix. As is it adds to the overall confusion in style and in context.
The descriptive statistics have been moved to an Appendix and the paper shortened in length.

Reviewer 2 Report
The author developed an interesting paper where he examines a mutual information theory approach linked to Qualia complexity, minimizing the exclusion and maximizes the interactions of the information system.
The manuscript presents novel results with respect to making decision theory and presented survey. However, some technical issues are not well presented. In order to following my review, I recommend to author to review the following issues:
L68: Please, fix the formula. The sum is from i=1 to M. Perhaps, word software produce this error. Use math editor in Word 2010.
L112: Denotes Mutual Information as MI. Also, define MI in terms of P2 L68 as:
MI[P,Q] = H[P] + H[Q] - H[PQ],
where Q is another discrete probability distribution (Arellano-Valle et al., 2013).
L167: I cannot see where is the link between proposed ITQ MI, and the another statistical tools used in the manuscript: correlations, kendall coefficients, ANOVA and Factor analysis. Perhaps, a Table or Figure about ITQ MI results is missing. I suggest to author to do a clear and concise section explaining the statistical methodologies used, and explaining the relationships between them (step by step).
L220: This must to be a Table, not a Figure.
Suggested Reference:
Arellano-Valle, R.B., Contreras-Reyes, J.E., Genton, M. G. (2013). Shannon Entropy and Mutual Information for Multivariate Skew‐Elliptical Distributions. Scandinavian Journal of Statistics, 40(1), 42-62.
Author Response
Reviewer Two Comments and Suggestions for Authors
The author developed an interesting paper where he examines a mutual information theory approach linked to Qualia complexity, minimizing the exclusion and maximizes the interactions of the information system. The manuscript presents novel results with respect to making decision theory and presented survey. However, some technical issues are not well presented. In order to following my review, I recommend to author to review the following issues:
Thank you for your kind remarks. I will endeavor to strengthen the technical issues listed below.
L68: Please, fix the formula. The sum is from i=1 to M. Perhaps, word software produce this error. Use math editor in Word 2010.
I have fixed the formula according to your suggestion. Given a time series, one of the most natural measures of disorder, and thus the absence of correlation, is Shannon entropy (Shannon and Weaver, 1949), which states that given a discrete probability distribution P = {p i : i = 1 to M}. Shannon entropy is defined as follows:
M
S[ P ] = − ∑ p i log (p i )
i =1
L112: Denotes Mutual Information as MI. Also, define MI in terms of P2 L68 as: MI[P,Q] = H[P] + H[Q] - H[PQ], where Q is another discrete probability distribution (Arellano-Valle et al., 2013).
The following statement has been added. The reduction of uncertainty can be quantified as mutual information (MI). MI can also be described in terms of P2 as MI[P,Q] = H[P] + H[Q] - H[PQ], where Q is another discrete probability distribution (Arellano-Valle et al., 2013).
L167: I cannot see where is the link between proposed ITQ MI, and the another statistical tools used in the manuscript: correlations, kendall coefficients, ANOVA and Factor analysis. Perhaps, a Table or Figure about ITQ MI results is missing. I suggest to author to do a clear and concise section explaining the statistical methodologies used, and explaining the relationships between them (step by step).
Table17. Qualia Consciousness and Information Theory Quale Integration with Organizational Competence
Table 17 summarizes the link between our proposed ITQ MI, and the statistical tools used in the manuscript: correlations, kendall coefficients, ANOVA and Factor analysis results. I have added a clear and concise section explaining the statistical methodologies used, and explained the relationships between them (step by step) as follows. ITQ is composed of five components. These are the state, uncertainty, mutual information integration and complexity of the collective consciousness. This decision making is contrasted with the individual consciousness or qualia components which include intrinsic, composition, information, integration and exclusion elements. We posit that under specific circumstances, constraints and assumptions the Entropy of the IS can be minimized and the organization performance of both individual and collective decision making improved. These specifics are addressed with the needed strongest of supports of increased IS trust, flexibility, business expertise, competitive advantage performance, and top management. Therefore, a regression analysis was run in order to demonstrate this potential relationship along with the individual contributions of these decision making supporters and their reliability for internal consistency purposes. The descriptive statistics and correlations supported these findings. Factor analysis was run in order to develop the supporters and demonstrate their interconstruct correlations. In order to compare and contrast individual and collective Entropy, Directional Measures such as Kendall Coefficients and Structured Equation Modeling were respectively employed to validate our model.
L220: This must to be a Table, not a Figure.
This has been changed to Table17. Qualia Consciousness and Information Theory Quale Integration with Organizational Competence
Suggested Reference:
Arellano-Valle, R.B., Contreras-Reyes, J.E., Genton, M. G. (2013). Shannon Entropy and Mutual Scandinavian Journal of Statistics, 40(1), 42-62.Information for Multivariate Skew‐Elliptical Distributions.
This suggested reference has been added to the literature review.
Round 2
Reviewer 1 Report
The authors did a good job in updating and correcting the text.
I would propose a minor addition to their references in the very first addition they made. Specifically in the sentence: "Most organizations' decision making takes place in the presence of conflict, paradox and noise [ADD REF]" .
This reference could be added
[ADD REF] = Nicolis, Gregoire and Basios, Vasileios, 2015
"Chaos, Information Processing and Paradoxical Games",
WORLD SCIENTIFIC, doi:10.1142/9145
Other than that all is well for publication.
Author Response
Reviewer One Comments and Suggestions for Authors
The authors did a good job in updating and correcting the text.
Thank you
I would propose a minor addition to their references in the very first addition they made. Specifically in the sentence: "Most organizations' decision making takes place in the presence of conflict, paradox and noise [ADD REF]" .
This reference could be added
[ADD REF] =
The following reference was added:
Nicolis, Gregoire and Basios, Vasileios, 2015
"Chaos, Information Processing and Paradoxical Games",
WORLD SCIENTIFIC, doi:10.1142/9145

Reviewer 2 Report
With respect to my last comments and suggestions, the author has done a good review
of the manuscript. For example, in lines 214-217 is defined the hypothesis of the study
based on proposed information quantifiers. However, some details are pending and I
recommend to review these parts for an acceptance recommendation:
1. L35: Delete here "Arellano-Valle et al. 2013", but keep this in L79.
2. L79: Include here the following sentence: "Let X and Y be two random variables
with discrete probability distributions P and Q, respectively; then, MI(P,Q)=0 when
P and Q are independent; otherwise, this index is positive (Cover & Thomas, 2006),
and it increases with the degree of dependence between the components pi and qi.
3. Put in legend of Figure 2 a more elaborated description of this scheme.
4. Put in legend of Table 1 a more elaborated description. For example: "this table includes
coefficient of determination (R), with their respective R Square and Adjusted Square". The
author could includes the description of predictors (bottom) in the legend. Also, fix the legend
of the rest of Figures in the same way: do not put footnote with letter "a", include this description
in the legend of the Tables.
5. L427: Table 11? This is the Figure 1? Why? Also, in P24 appears this Figure. This figure is
already in the body of the manuscript. Delete this.
6. L429-447: Why you explain in the manuscript in 1rst person? Please, revise the redaction
of this paragraph. Also, perhaps this description could be presented in the beginning of
methodologies, not in Results.
7. L450: The legend is missing. Also, revise title of this plot.
8. L497: Delete this title. This paragraph is part of Discussion.
9. L527: I am agree with the author about limitations of Shannon entropy. The author could
include a sentence describing a further work using Mutual information of Rényi entropy
(Contreras-Reyes, 2015; Eggels and Crommelin, 2019).
10. L576 & 579: Where are mentioned the Appendices A and B in the body of manuscript?
Perhaps, this could be a supplementary file?
11. L680-682: Fix these lines as: "Shannon Entropy and Mutual Information for Multivariate
Skew-Elliptical Distributions. Scandinavian Journal of Statistics, 40(1), 42-62".
Suggested References:
Cover, T.M., Thomas, A.J. (2006). Elements of information theory. 2nd edition.
Willey-Interscience, NJ, USA.
Contreras-Reyes, J.E. (2015). Rényi entropy and complexity measure for skew-gaussian
distributions and related families. Physica A 433, 84-91.
Eggels, A., Crommelin, D. (2019). Quantifying Data Dependencies with Rényi Mutual
Information and Minimum Spanning Trees. Entropy 21, 100.
Author Response
Reviewer Two Comments and Suggestions for Authors
With respect to my last comments and suggestions, the author has done a good review
of the manuscript. For example, in lines 214-217 is defined the hypothesis of the study
based on proposed information quantifiers. However, some details are pending and I
recommend to review these parts for an acceptance recommendation:
1. L35: Delete here "Arellano-Valle et al. 2013", but keep this in L79.
Deleted
2. L79: Include here the following sentence: "Let X and Y be two random variables
with discrete probability distributions P and Q, respectively; then, MI(P,Q)=0 when
P and Q are independent; otherwise, this index is positive (Cover & Thomas, 2006),
and it increases with the degree of dependence between the components pi and qi.
Sentence has been added.
3. Put in legend of Figure 2 a more elaborated description of this scheme.
Figure 2. Testing of our Research Model hypotheses using SEM
4. Put in legend of Table 1 a more elaborated description. For example: "this table includes
coefficient of determination (R), with their respective R Square and Adjusted Square". The
author could includes the description of predictors (bottom) in the legend. Also, fix the legend
of the rest of Figures in the same way: do not put footnote with letter "a", include this description
in the legend of the Tables.
Legends added
5. L427: Table 11? This is the Figure 1? Why? Also, in P24 appears this Figure. This figure is
already in the body of the manuscript. Delete this.
Table Deleted
6. L429-447: Why you explain in the manuscript in 1rst person? Please, revise the redaction
of this paragraph. Also, perhaps this description could be presented in the beginning of
methodologies, not in Results.
Redacted and moved to methodologies section
7. L450: The legend is missing. Also, revise title of this plot.
Figure 4 Scree Plot
8. L497: Delete this title. This paragraph is part of Discussion.
Deleted Limitations title.
9. L527: I am agree with the author about limitations of Shannon entropy. The author could
include a sentence describing a further work using Mutual information of Rényi entropy
(Contreras-Reyes, 2015; Eggels and Crommelin, 2019).
While the most common way of measuring information is through the Shannon entropy lens, there are others such as Rényi entropy, that was developed by the Hungarian mathematician Alfréd Rényi. It generalizes Shannon entropy and includes other entropy measures as special cases ((Contreras-Reyes, 2015; Eggels and Crommelin, 2019; Cover 2006).
10. L576 & 579: Where are mentioned the Appendices A and B in the body of manuscript?
Perhaps, this could be a supplementary file?
Appendices A and B removed from body of manuscript
11. L680-682: Fix these lines as: "Shannon Entropy and Mutual Information for Multivariate
Skew-Elliptical Distributions. Scandinavian Journal of Statistics, 40(1), 42-62".
Line has been fixed.
Suggested References:
Cover, T.M., Thomas, A.J. (2006). Elements of information theory. 2nd edition.
Willey-Interscience, NJ, USA.
Contreras-Reyes, J.E. (2015). Rényi entropy and complexity measure for skew-gaussian
distributions and related families. Physica A 433, 84-91.
Eggels, A., Crommelin, D. (2019). Quantifying Data Dependencies with Rényi Mutual
Information and Minimum Spanning Trees. Entropy 21, 100.
References added.
Round 3
Reviewer 2 Report
After two rounds of revision, the author has done a good review of the manuscript.Thanks for considering my comments and suggestions.